# Ask Doctor Smartphone! An App to Help Physicians Manage Foreign Body Ingestions in Children

**DOI:** 10.3390/diagnostics13203285

**Published:** 2023-10-23

**Authors:** Marco Di Mitri, Giovanni Parente, Cristian Bisanti, Eduje Thomas, Sara Maria Cravano, Chiara Cordola, Marzia Vastano, Edoardo Collautti, Annalisa Di Carmine, Michela Maffi, Simone D’Antonio, Michele Libri, Tommaso Gargano, Mario Lima

**Affiliations:** Pediatric Surgery Department, IRCCS Sant’Orsola-Malpighi Polyclinic, Alma Mater Studiorum—University of Bologna, 40138 Bologna, Italy; giovanni.parente@outlook.com (G.P.); bisanticristian96@gmail.com (C.B.); edu.thomas92@gmail.com (E.T.); sara-cravano@libero.it (S.M.C.); chiaramberle@gmail.com (C.C.); marzia.vastano@icloud.com (M.V.); edocolla.ec@gmail.com (E.C.); annalisa.dicarmine@gmail.com (A.D.C.); michela.maffi@libero.it (M.M.); simone.dantonio@aosp.bo.it (S.D.); michele.libri@aosp.bo.it (M.L.); tommaso.gargano2@unibo.it (T.G.); mario.lima@unibo.it (M.L.)

**Keywords:** foreign body ingestion, foreign body removal, GI endoscopy, upper endoscopy, pediatric endoscopy, smartphone, app, Apple, Android, health promotion

## Abstract

Background: Foreign body ingestion (FBI) represents the most common cause of emergent gastrointestinal endoscopy in children. FBI’s management can be quite challenging for physicians because of the variability of the clinical presentation, and the decision tree becomes even more intricate because of patient-specific variables that must be considered in the pediatric age range (e.g., age of patients and neuropsychiatric disorders) in addition to the mere characteristics of the foreign body. We present an application for smartphones designed for pediatricians and pediatric surgeons based on the latest guidelines from the official pediatric societies. The app aims to help physicians manage FBI quickly and properly in children. Materials and methods: The latest pediatric FBI management guidelines were reviewed and summarized. The flow chart we obtained guided the development of a smartphone application. A questionnaire was administered to all pediatric surgeon trainees at our institute to test the feasibility and helpfulness of the application. Results: An app for smartphones was obtained and shared for free on the Google Play Store and Apple Store. The app guides the physician step by step in the diagnostic process, analyzing all patient- and foreign body-specific characteristics. The app consultation ends with a suggestion of the most proper decision to make in terms of further radiological investigations and the indication and timing of endoscopy. A questionnaire administered to trainees proved the app to be useful and easy to use. Conclusion: We developed an app able to help pediatricians and pediatric surgeons manage FBI in children, providing standardized and updated recommendations in a smart and easily available way.

## 1. Introduction

Foreign body ingestion (FBI) represents the most common cause of emergent gastrointestinal (GI) endoscopy in children. In determining a variety of different clinical manifestations, from asymptomatic to life-threatening events, FBI’s management can be quite challenging for physicians. Furthermore, the decision tree becomes even more intricate because of patient-specific variables that must be considered in the pediatric age range (e.g., age of patients, neuropsychiatric disorders) in addition to the mere characteristics of the foreign body.

Up to 75% of the total ingestions usually occur in 5-year-old children or younger [1].

Contrarily to adults, about 98% of FBI in children is accidental and involves common objects found at home, such as coins, toys, jewels, magnets, and button batteries [2].

Although most swallowed foreign bodies (FB) pass the GI tract without any consequence, in 10–20% of cases, an endoscopic intervention is necessary to avoid complications, such as poisoning, impaction, ulceration and bleeding, perforation, and, in the most severe cases, death [3,4,5,6].

In most cases, a detailed history along with a physical examination are enough for a diagnosis. Radiopaque FBs benefit from a chest-abdomen X-ray (XR) that will confirm the ingestion and provide the exact position. When complained, indirect signs of ingestion suggest the FB position (e.g., esophagus in the case of sialorrhea) or ongoing complication (e.g., acute abdomen, vomit, hematemesis, etc.). Selected cases require further investigations such as a contrast XR (e.g., radiolucent FBs) or a CT scan (e.g., suspicion of an aorto-esophageal fistula, intestinal perforation, etc.).

In the diagnostic process, there are several other features that must be considered: patient-specific characteristics such as age and comorbidities (e.g., inflammatory bowel disease, intestinal stenosis, and autism), as well as FB-specific characteristics such as dimension, shape, and potential for developing complications (e.g., disk batteries and magnets) [7].

In general terms, endoscopic removal is warranted for a FB stuck in the esophagus, just as dangerous or voluminous FBIs in the stomach require endoscopic removal. Patients with a FB that has passed the duodenum can be discharged, asking parents to look at the feces to find out the ingested object, or admitted for clinical observation and a scheduled XR in cases of dangerous FBs.

Undeniably, inappropriate management of FBI in children can lead to serious short- and long-term complications [8,9,10,11]. To avoid the latter, it is mandatory for pediatricians and pediatric surgeons to know the perfect timing for endoscopic FB removal.

Nowadays, the widespread use of smartphones and their easy connection to the internet have determined the development and multiplication of applications that support physicians in their daily practice.

We revised the latest guidelines in terms of FBI in children, approved by the most important societies worldwide (NASPGHAN, North American Society for Pediatric Gastroenterology Hepatology and Nutrition, and ESPGHAN, European Society for Pediatric Gastroenterology Hepatology and Nutrition) [2], and, along with our experience, we summarized the recommendations in an application for smartphones that shows our proposed diagnostic-therapeutic algorithm. The application is available for free on the Google Play Store and the Apple Store.

In this paper, we present a summary of the reviewed guidelines that allowed the development of the smartphone application, which is then described to encourage its diffusion.

## 2. Methods

### 2.1. App Design

The latest guidelines for the management of FBI in children were reviewed. Such evidence was coupled with the experience of our team of pediatric surgeons who perform GI endoscopy, and a flowchart summarizing the diagnostic-therapeutic algorithm of each type of FB was outlined [2,3,12].

In collaboration with StdOut S.r.l. (Palermo, Italy), these flowcharts guided the development of an application for smartphones called “Foreign bodies”, available for free on the Google Play Store and the Apple Store.

### 2.2. App Evaluation

Trainees of our pediatric surgical department were provided with the FBI App for a test period of about one year (January 2022–December 2022). Trainees were asked to make their own diagnostic/therapeutic decision first. Furthermore, they were allowed to verify the proposed flow chart on the application. Finally, trainees were recommended to take note of any concordances or discrepancies between what they initially thought and what was suggested by the application, as well as to report their impressions on the FBI app.

At the end of the test period, trainees were asked to answer anonymously to the following questions:(1)Was the app consultation easy and intuitive?(2)Was there an agreement between your diagnostic/therapeutic approach and the ones suggested by the app?(3)Was your decision-making process faster with the help of the app?(4)Did the app consider every characteristic of the patient when elaborating the proper diagnostic/therapeutic flow?(5)Did you experience an improvement in patient management?(6)Have you ever thought you would suggest the app to a colleague?

The possible answers to the questions were one of the following: never, almost never, sometimes, often, almost always.

## 3. Results

### 3.1. Management of FBI: Summary of the Latest Guidelines

In regard to the diagnostic and therapeutic flowcharts in the application, we propose the following management:

#### 3.1.1. Esophageal Food Impaction

Esophageal food impaction (EFI) in children is mostly secondary to comorbidities, such as anastomotic stenosis after esophageal atresia repair, eosinophilic esophagitis (EoE), reflux esophagitis, or, rarely, achalasia [13,14]. The most common symptoms are sialorrhea, dysphagia, odynophagia, and epigastric pain. Emergent endoscopic removal is required in cases of absent spontaneous clearance and symptomatic patients; on the contrary, if patients can swallow saliva, the endoscopy may be delayed up to 24 h. If an EFI is diagnosed in a patient without known comorbidities, esophageal biopsies should be taken into account to exclude EoE (during the same procedure or delayed) [15,16,17,18,19]. The use of glucagon to hasten spontaneous clearance is not recommended [3,20,21] (Figure 1).

#### 3.1.2. Sharp Objects

Pointed foreign bodies are typically described as sharp objects and include nails, pins, tacks, toothpicks, and much more. Symptoms are likely to arise when they get stuck in the upper-mid esophagus, with epigastric pain, dysphagia, and sialorrhea being the most prominent symptoms.

However, up to 50% of cases may be asymptomatic, even in cases of proximal intestinal perforation [22]. Complications include perforation and extraluminal migration, abscess, mediastinitis or peritonitis, fistula formation, appendicitis, and extraluminal migration (liver, bladder, heart, and lung penetration) [23,24,25].

The ileocecal region is the most common site of intestinal perforation [26]. The most controversial aspect of the management of a vulnerating FBIs is related to the definition of a vulnerating FB itself. Nevertheless, we propose the emergent endoscopic removal of every sharp FB in the esophagus and stomach. In case the FB is passing the duodenum, we suggest hospitalization and seriated XRs both to observe the progression of the FB and to promptly detect signs of complications (Figure 2).

#### 3.1.3. Superabsorbent Objects

Superabsorbent objects are made of polymers that, in contact with water, increase their volume, reaching up to 100 times their original dimension. In cases of ingestion, an emergency endoscopic removal is recommended. If the FB is beyond the stomach, hospitalization is recommended to promptly recognize an intestinal obstruction [2,27] (Figure 2).

#### 3.1.4. Blunt Objects

Conventionally considered as not sharp objects, disk batteries, magnets (when more than one), or superabsorbent. Coins are the most commonly swallowed FBs by children. Generally, spontaneous clearance of coins occurs in approximately 30% of patients, depending on the size of the coin and the age of the patient. Considering data reported in the literature, the dimension of the FB is important too, since objects with a diameter >24–25 mm, especially in children younger than 5 years old, or particularly long (>6 cm), are unlikely to pass the pylorus and should be removed even if blunt (but do not represent an emergency) [2,5,6,28]. Asymptomatic patients who ingested a not-vulnerable FBI can be discharged by asking parents to examine the feces, looking for the ingested object (Figure 3).

#### 3.1.5. Multiple Magnets

If a single magnet can be considered a blunt foreign body, the ingestion of multiple magnets may result in tremendous complications. The small size and shiny aspect of these magnets make them an attractive target for ingestion by infants and toddlers. The main risk of this event is the potential entero-enteric fistula formation between magnets in adjacent bowel loops, which entails bowel perforation, ischemia, necrosis, and peritonitis [3]. Patients who swallow multiple magnets must be subjected to emergent endoscopy if the FB is in the stomach; if magnets pass beyond the stomach, hospitalization and scheduled XRs are recommended to early detect signs of complications (Figure 4).

#### 3.1.6. Disk Battery (BB)

In regard to BB ingestion, several mechanisms of injury have been theorized, such as direct electrical discharge, local pressure necrosis, corrosive damage from leakage of the BB’s contents, and heavy metal poisoning [11]. Thus, emergent endoscopic removal is required. Fortunately, BBs that pass through the esophagus often proceed through the GI tract without any consequences. The risk of the occurrence of an aorto-esophageal fistula deserves a particular mention: when suspected, no BB removal should be attempted without the presence of a heart surgeon [2,26] (Figure 5).

### 3.2. App Development

In collaboration with StdOut S.r.l., we developed a hybrid mobile app with an ionic capacitor framework based on compatible cross-platforms with a unique code base.

“Foreign bodies” APP is free and does not require user registration, nor does it manage or save user data, guaranteeing security and privacy.

We divided the categories of foreign bodies into five groups: disk batteries, magnets, food boluses, sharp and superabsorbent objects, and blunt objects. After clicking on the category of interest, we breeze through a series of screens in which we are asked to provide details about all possible variables, such as the patient’s age, dimension of the FB, time from ingestion, symptoms, and comorbidities. At last, we enter a final screen showing the correct preliminary management through first-level diagnostic exams based on our previous selections (Figure 6).

Through the main menu, it is also possible to contact our center for any supplementary information about the management of FBI.

### 3.3. Questionnaire

All the trainees (10 out of 10) answered the administered questionnaire.

Results are summarized in Table 1 and illustrated in Figure 7.

## 4. Discussion

FBI is one of the most common causes of childhood referrals to pediatric emergency departments worldwide. Presenting with an extremely heterogeneous spectrum of clinical manifestations, from asymptomatic to life-threatening, FBI may be challenging in terms of diagnosis and therapeutic decisions. Indeed, several features should be taken into account when choosing the proper treatment.

Surely, FBI should be considered a time-related condition. Thus, the faster the decision is made, the lower the potential complications. As a matter of fact, inaccurate knowledge of the most recent guidelines, slow decision-making, and the absence of specialized physicians (e.g., pediatric endoscopists) in the first patient assessment are reported to be the main risk factors of a wrong and/or delayed diagnostic-therapeutic process, which can result in severe short- and long-term complications.

One of the most prominent shifts within the last two decades was the explosive embrace of smartphones. The release of the first-generation iPhone (Apple Technology, Cupertino, CA, USA) in 2007 (as well as its Android competition shortly thereafter) quickly led to its widespread public adoption. The surprising success of the iPhone, however, came not only through a different design but also because of a different competition strategy [29]. Apple created the App Store to make shopping for apps convenient and engaging while also building a community of developers to keep the App Store brimming with options. This meant that the company was able to create a whole business model around applications that made their phones infinitely customizable. The number of smartphone owners has grown exponentially since its introduction, and today there are 4.3 billion smartphones globally [30]. This ubiquity has resulted in their use in nearly all aspects of life, and clinical practice is no exception.

According to recent surveys, there are currently about 325,000 smartphone apps available on health-related topics, and their usefulness is undergoing evaluation by various authors [31].

Rubin et al. evaluated the adhesion to guidelines in the management and treatment of stroke in a group of residents compared with another that used a specific app; the latter showed a major and statistically significant adherence to guidelines [32].

Nason et al. administered a questionnaire to all urology trainees in Ireland, investigating whether they downloaded medical apps, the frequency of use, and their opinion. 77.8% of the respondents reported a daily use of smartphone applications designed for urologists, and 87% found them a useful tool in their daily practice [33].

In a systematic review, Prgomet et al. analyzed the impact of smartphone apps on the daily practice of healthcare providers. They conclude that medical apps positively affect the speed and accuracy of diagnosis, error prevention, information accessibility, and data management in healthcare settings [34]. Kernebeck et al., in their review on the same topic, even if limited to the gastroenterological field, concluded that the app consultation improved the diagnosis’ accuracy, treatment choice, and follow-up of chronic diseases [35].

The extremely encouraging data reported above inspired our group to review and summarize the guidelines of FBI in children. Furthermore, we included the results in a brand-new application for smartphones entirely dedicated to FBI.

This product provides colleagues with the latest diagnostic and therapeutic algorithms, always ready to be consulted in their pockets. To make sure it can always be accessible even if an internet connection is not available, the application comes with an offline mode.

Moreover, being in a third-level center with renowned experience in endoscopy and its indications in FBI, we included in the app the possibility to contact us for further clarifications on this topic.

We believe this app can improve diagnostic performance by reducing the number of unnecessary endoscopies and XRs. Moreover, we are convinced that it can be a precious tool for pediatricians in peripheric pediatric emergency departments, helping them decide which patients should be referred to a third-level center and which may be discharged.

Finally, by sharing such a fast and easy way to have access to guidelines on FBI, we strongly believe we can reduce the number of misdiagnosis and complications.

Clearly, guidelines evolve rapidly, so the app requires regular updates.

In regard to the application satisfaction questionnaire given to the trainees (Figure 7 and Table 1), we can infer that the app was easy and intuitive to use. Moreover, the feedback from the respondents appraises the comprehensiveness of the application as it takes into account every patient characteristic for a proper and customized diagnostic-therapeutic process. The questionnaire showed a remarkable 20% of trainees declaring that the original diagnostic–therapeutic protocol was different from the app’s proposal. In these cases, the use of the application saved trainees from misdiagnosis or mistreatment. In particular, the cases of mismatch between colleagues’ ideas and app outcomes were: an ingestion of DB at least 3 h before access to the emergency department where the trainee did not consider a CT angiography (Section 3.1.6 and Figure 5); a single-magnet ingestion (in the stomach at the XR) in a patient affected by autism where endoscopic removal was originally not considered (see Section 3.1.5 and Figure 4); a case of blunt screw ingestion (in the small bowel at the XR) where the app suggested discharge, whereas the original program of the trainee would have been admission in a pediatric surgery department for observation.

As suggested by trainees’ answers, the app speeded up the diagnostic/therapeutic process, saving precious time in the emergency department.

Finally, they reported a positive effect on the quality of patients’ management and would recommend the app’s adoption to other colleagues.

## 5. Limits of the Study

First, as guidelines are regularly updated, our app must be kept on track. Moreover, even the smartphones’ informatic systems are regularly updated, which has a great influence on the compatibility of the app with new mobile models. Therefore, we know we must continuously dedicate efforts to updating the app, and this requires dedicated healthcare providers.

Second, even if the app is freely available on platforms accessible to everyone, its usage should be reserved for physicians only. We are currently studying a way to verify the user identity to discourage patient usage (we believe a parent should always seek medical evaluation in the case of FBI).

Moreover, we are aware of the vastness of the field of FBI ingestion in children. Therefore, certainly some difficult or rare cases would not find an appropriate allocation in the decision tree of the app, preventing it from giving a proper decision outcome. For the future, we are considering the possibility of powering the app through AI and machine learning. However, we are sure that our app covers the majority of common cases physicians can encounter in the emergency department, and, regardless of any app or guidelines suggestions, rare cases should always be discussed with expert pediatric endoscopists. We therefore do not consider rare or difficult cases an important drawback of our app.

Finally, we know that the number of clinicians with whom we tested the app is too low to assess content and construct validity. Our main aim was to present our preliminary results. It would be interesting to spread the application to a larger number of colleagues and collect their impressions to speculate about the real usefulness in daily clinical practice of this application.

## 6. Conclusions

Our foreign body smartphone application simplifies and speeds up access to guidelines, providing standardized and updated indications for the management of FBI.

Data confirming the positive effects of this app on diagnosis and decision-making should be recorded to promote its diffusion globally.

## Figures and Tables

**Figure 1 diagnostics-13-03285-f001:**
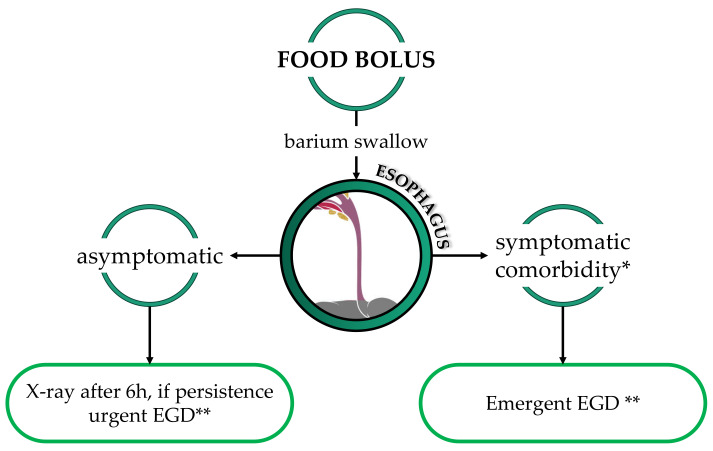
Diagnostic and therapeutic flowchart in the case of food bolus. * Comorbidities: known esophageal stenosis, history of esophageal atresia, ** Urgent EGD: to be performed within 24 h; emergent EGD: to be performed within 2 h.

**Figure 2 diagnostics-13-03285-f002:**
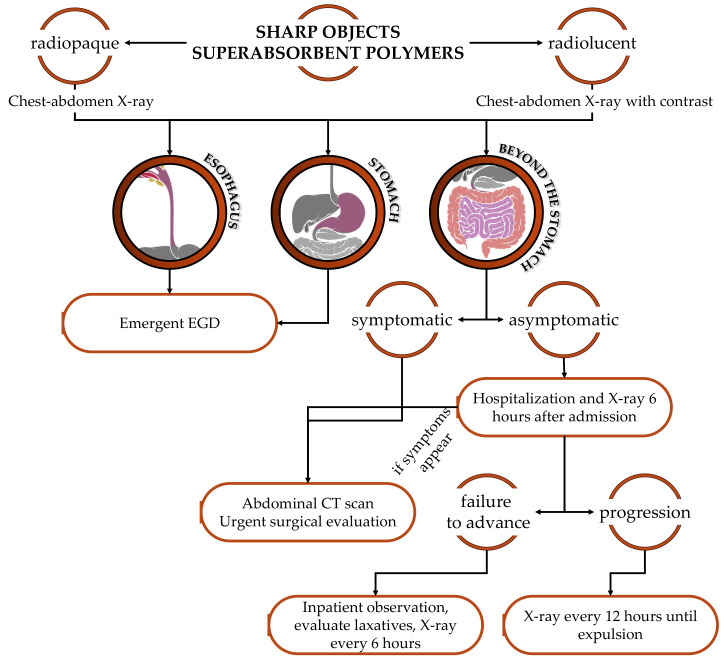
Diagnostic and therapeutic flowchart in case of sharp objects/superabsorbent polymers ingestion. Urgent EGD: to be performed within 24 h; emergent EGD: to be performed within 2 h.

**Figure 3 diagnostics-13-03285-f003:**
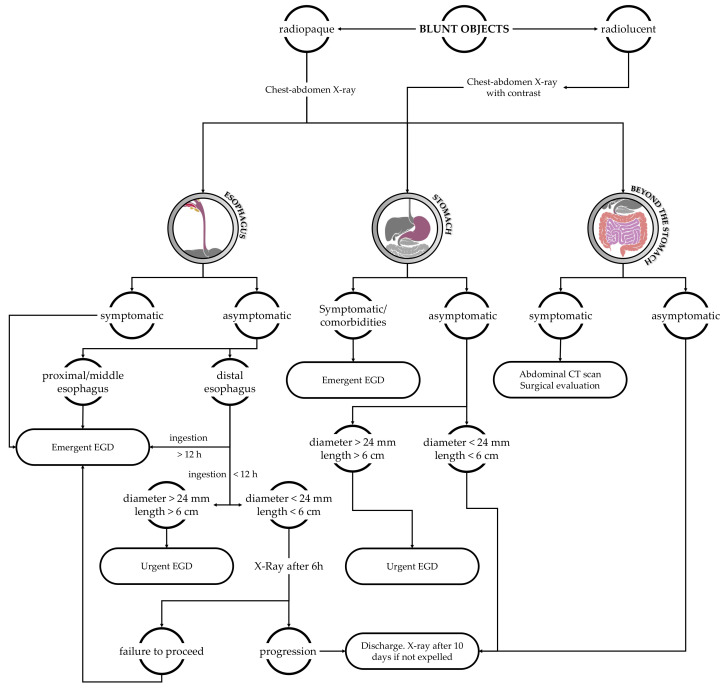
Diagnostic and therapeutic flowchart in the case of blunt object ingestion. Urgent EGD: to be performed within 24 h; emergent EGD: to be performed within 2 h.

**Figure 4 diagnostics-13-03285-f004:**
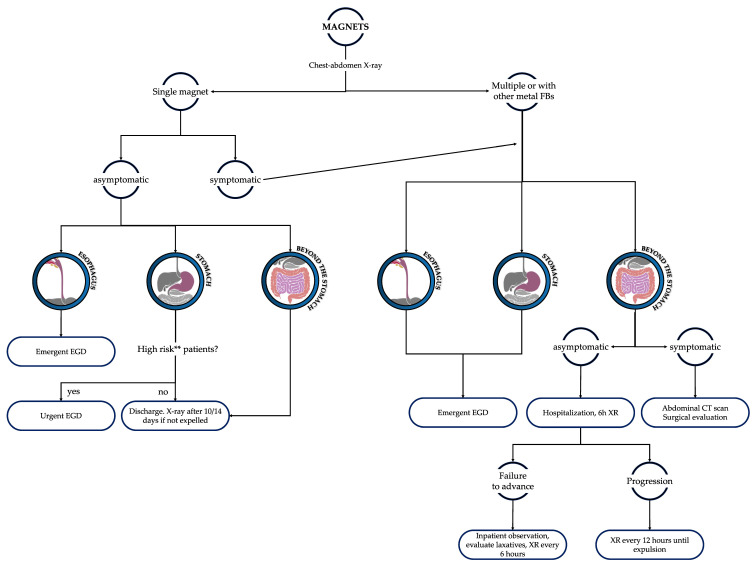
Diagnostic and therapeutic flowchart in the case of magnet ingestion. Urgent EGD: to be performed within 24 h; emergent EGD: to be performed within 2 h. ** High-risk patients: autism or other comorbidities that increase the risk of ingestion of other FBs (such as other magnets or metal FBs).

**Figure 5 diagnostics-13-03285-f005:**
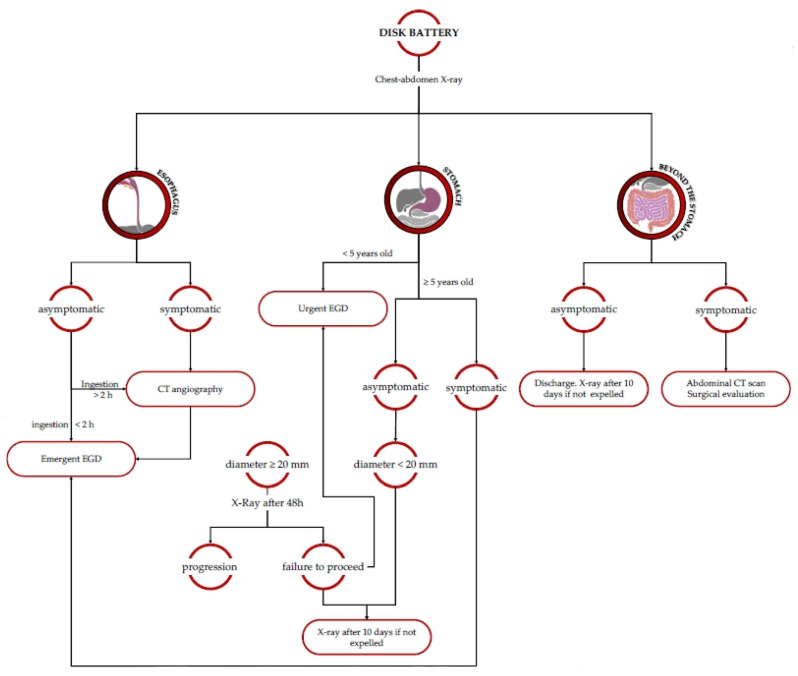
Diagnostic and therapeutic flowchart in the case of disk battery ingestion. Urgent EGD: to be performed within 24 h; emergent EGD: to be performed within 2 h.

**Figure 6 diagnostics-13-03285-f006:**
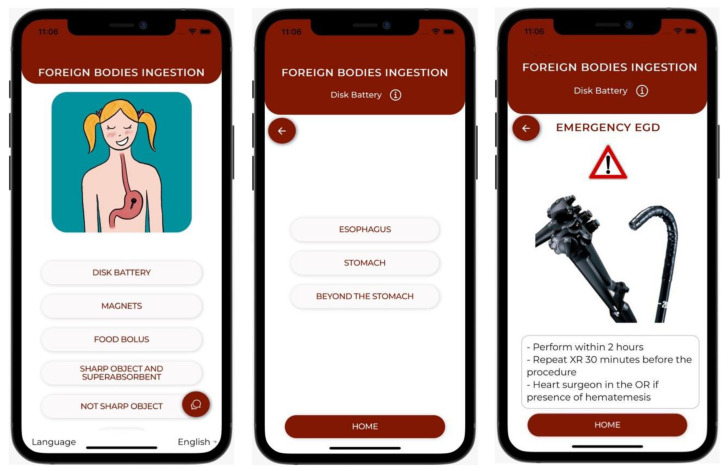
Examples of app interfaces. Main menu and following screens.

**Figure 7 diagnostics-13-03285-f007:**
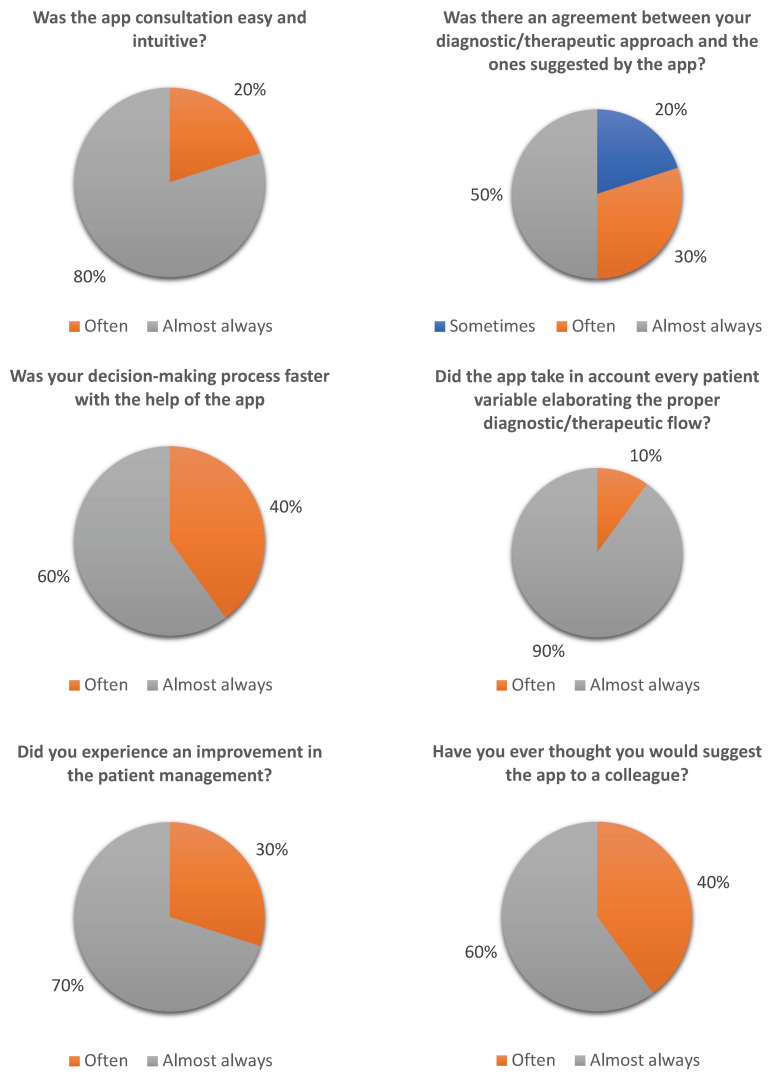
Graphic representation of trainees’ satisfaction questionnaire.

**Table 1 diagnostics-13-03285-t001:** Answers to the questionnaire given to trainees (*n* = 10).

	NEVER	ALMOST NEVER	SOMETIMES	OFTEN	ALMOST ALWAYS
(1) Was the app consultation easy and intuitive?	0	0	0	2	8
	(0%)	(0%)	(0%)	(20%)	(80%)
(2) Was there an agreement between your diagnostic/therapeutic approach and the ones suggested by the app?	0	0	2	3	5
	(0%)	(0%)	(20%)	(30%)	(50%)
(3) Was your decision-making process faster with the help of the app?	0	0	0	4	6
	(0%)	(0%)	(0%)	(40%)	(60%)
(4) Did the app take into account every patient variable, elaborating the proper diagnostic/therapeutic flow?	0	0	0	1	9
	(0%)	(0%)	(0%)	(10%)	(90%)
(5) Did you experience an improvement in patient management?	0	0	0	3	7
	(0%)	(0%)	(0%)	(30%)	(70%)
(6) Have you ever thought you would suggest the app to a colleague?	0	0	0	4	6
	(0%)	(0%)	(0%)	(40%)	(60%)

## Data Availability

Data is available upon reasonable request from the author.

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
