# Peer review of "Ask Doctor Smartphone! An App to Help Physicians Manage Foreign Body Ingestions in Children"

_diagnostics, 2023, doi:10.3390/diagnostics13203285_

Round 1
Reviewer 1 Report
I have read the manuscript with greatest interest the manuscript. I have no concerns for further publication
Author Response
Dear Reviewer,
Thank you for your comments.
Reviewer 2 Report
This work presenting and discussing an app named Foreign Bodies
to manage foreign body ingestions in children. The app follows decision tree rules to guide the diagnostic-therapeutic process. The paper is well-written. However, the following issues hindering the scientific contribution:
Comments:
-These decision tree rules deriving the outcomes are static
and don't exhibit lifelong learning.
-It is not an AI-based app deriving the outcome. Moreover,
the search space is huge for such decision trees and the study does not discuss how to automatically handle this issue ,hindering the app
-Suppose the case where these decision trees might not be of interest (or related) to the users. In that case, what would be the behavior of the app?
That is my primary objection
Author Response
Dear Reviewer,
Thank you for your precious comments that enhanced the scientific quality of our paper.
In the following lines I will answer to your kind requests.
[…]
-These decision tree rules deriving the outcomes are static and don't exhibit lifelong learning.
It is true that the algorithm on which the app is based is quite static. However, it resembles the guidelines of the most important societies of pediatric gastroenteroloy in conjunction with our experience of more than 50 years of pediatric surgeons dedicated to pediatric endoscopy.
This concept is now better specified in the text.
-It is not an AI-based app deriving the outcome. Moreover, the search space is huge for such decision trees and the study does not discuss how to automatically handle this issue, hindering the app
The application of AI to this topic and, consequently, to our app is an extremely interesting suggestion that we are now considering for the future.
Although it is true that the field of FBI is huge and difficult to enclose entirely in rigid decision trees, as said before, we added to the guidelines our personal experience enlarging the possibility of success of our help in managing FBI episodes in children.
We are aware that there will be cases not covered by our app, but we are sure they will be rare.
Moreover, difficult cases should always be discussed with expert pediatric endoscopists regardless any app or guidelines suggestion.
-Suppose the case where these decision trees might not be of interest (or related) to the users. In that case, what would be the behavior of the app?
As said in the answer of your previous kind comment, our app covers the majority of FB and patient variabilities. However, users will certainty encounter cases in which the app cannot help. In this case the app decision tree will stop because the users will not find the item of interest during the decision tree.
The latter situation belongs to rare or difficult case that should always be discussed with pediatric endoscopists. We therefore think this could not be an important point against our app that is designed to help physicians in the most common cases.
It is now better specified in the text.
We hope we answered to all your kind comments, and we thank you a lot again for your suggestions.
Reviewer 3 Report
Dear authors,
I received your submission as a reviewer and found that you have tried to introduce your new released app for guiding the management of foreign body ingestion in children. It is an attractive work and would definitely helpful, if it could be included in one of current medical apps in terms of patient management in various fields. As you know there are considerable number of comprehensive apps in this regard.
However, when it comes to medical writing and journalism, there are approved structures that should be respected, and the current text of your submission is not compatible with. If you intend to introduce your app, it is suitable to write a “Letter to the editor” and tell your story about the process of its design, and also its novelty or superiority. Or, you can prepare a “Review article” on latest guidelines of foreign body ingestion in children and tell how you classified them to a unique algorithm and how did you revise their limitations by merging them together, and how exactly you did it. Or, present an “Original article” on how you assessed the accuracy, content validity or etc. of your app with all details as a useful medical paper for the usual audiences.
In my opinion, the current text is not compatible with any of mentioned structures, so I cannot recommend it for publication. I highly recommend you to re-write the whole paper with a clear aim and pre-defined structure.
Kind regards,
Author Response
Dear Reviewer,
Thank you for your precious comments that enhanced the scientific quality of our paper.
In the following lines I will answer to your kind requests.
[…]
However, when it comes to medical writing and journalism, there are approved structures that should be respected, and the current text of your submission is not compatible with. If you intend to introduce your app, it is suitable to write a “Letter to the editor” and tell your story about the process of its design, and also its novelty or superiority. Or, you can prepare a “Review article” on latest guidelines of foreign body ingestion in children and tell how you classified them to a unique algorithm and how did you revise their limitations by merging them together, and how exactly you did it. Or, present an “Original article” on how you assessed the accuracy, content validity or etc. of your app with all details as a useful medical paper for the usual audiences.
In my opinion, the current text is not compatible with any of mentioned structures, so I cannot recommend it for publication. I highly recommend you to re-write the whole paper with a clear aim and pre-defined structure.
We completely agree that, in order to be a solid "original article", a more structured questionnaire subjected to a larger number of physicians is necessary to obtain significant content and construct validity.
Nevertheless, we tested the app with our trainees and, even if poor in terms of numerosity, it represents an interesting preliminary result to be discussed with scientific community in our opinion.
What just said is better discussed in the new limits section of our revised paper.
However, we understand your kind and punctual objection and we will surely change the paper format according to editorial suggestions.
We hope we answered to all your kind comments, and we thank you a lot again for your suggestions.
Reviewer 4 Report
I believe the app's content is quite interesting and can be of great value to newly graduated MDs.
But I would like to see a bigger group testing it. Only 10 students are too little. Can you increase the number of test users to at least 30?
Moreover, can you explore the results from the questionnaire in the discussion? You spend a lot of time talking about the use of smartphones and apple (3rd paragraph in the discussion) which is not really the focus of the paper and I believe to not be necessary. In fact, I would remove most of that text.
Can you specifically explore question 2 in your discussion? How do you explain that for 20% of the users, the diagnostic was only in agreement sometimes? Was it a misdiagnosis of the ap or the MD?
Finally, for the last 2 questions, the results are based on a subjective evaluation regarding the app value. Can you compare the performance of newly graduated MDs using the app and not using it? To have more direct measurements of its benefit. It would be nice to see such a comparison incorporated into the manuscript.
I would review the text for cadence. The portions where you explain the different types of FBIs are a bit slow to read.
Author Response
Dear Reviewer,
Thank you for your precious comments that enhanced the scientific quality of our paper.
In the following lines I will answer to your kind requests.
[…] I would like to see a bigger group testing it. Only 10 students are too little. Can you increase the number of test users to at least 30?
Moreover, can you explore the results from the questionnaire in the discussion? You spend a lot of time talking about the use of smartphones and apple (3rd paragraph in the discussion) which is not really the focus of the paper and I believe to not be necessary. In fact, I would remove most of that text.
Can you specifically explore question 2 in your discussion? How do you explain that for 20% of the users, the diagnostic was only in agreement sometimes? Was it a misdiagnosis of the ap or the MD?
Finally, for the last 2 questions, the results are based on a subjective evaluation regarding the app value. Can you compare the performance of newly graduated MDs using the app and not using it? To have more direct measurements of its benefit. It would be nice to see such a comparison incorporated into the manuscript.
We perfectly agree that to obtain a more reliable construct and content validity of our app we need larger numerosity of clinicians involved in the satisfaction questionnaire.
Unfortunately, it requires a lot of time (at least a year to have sufficient cases for each colleague involved) therefore we are sorry we cannot satisfy your kind and proper request.
Nonetheless, we know that our results could be defined nothing more than preliminary; this is why we are currently testing the app to a larger population of colleagues, and we hope to collect results soon.
What just said is reported in our new limits of the study section in the revised paper.
Finally, we dedicated some lines more to the discussion of the questionnaire as you suggested briefly presenting the cases you asked for.
We are sorry that other parts in the discussion section were too wordy, but they were an editorial request.
We performed review by a native English speaker.
We hope we answered to all your kind comments, and we thank you a lot again for your suggestions.
Round 2
Reviewer 2 Report
Authors justified raised comments in the previous round of review.
Minor Comments:
In line 254,
CHANGE
(fig.7 and tab.1)
TO
(Figure 7 and Table 1)
Note: first letter of words (Figure, Table) has to be in upper case throughout the manuscript. Spell out words (e.g., changing Tab. to Table) in line 197
Reviewer 3 Report
Dear authors,
I received your revised version. Despite of conducted revisions, the methods still suffers from important limitations in terms of validity as you mentioned in your reply. In my viewpoint, the paper is not qualified to be an original article, and just would be appropriate as a "letter to the editor".
Kind regards,
Reviewer 4 Report
I appreciate your reply and I am looking forward to see future studies with a bigger N once you complete the tests.